# Analysis of Chromatin Accessibility and DNA Methylation to Reveal the Functions of Epigenetic Modifications in *Cyprinus carpio* Gonads

**DOI:** 10.3390/ijms25010321

**Published:** 2023-12-25

**Authors:** Mingxi Hou, Qi Wang, Ran Zhao, Yiming Cao, Jin Zhang, Xiaoqing Sun, Shuangting Yu, Kaikuo Wang, Yingjie Chen, Yan Zhang, Jiongtang Li

**Affiliations:** 1Key Laboratory of Aquatic Genomics, Ministry of Agriculture and Rural Affairs, Beijing Key Laboratory of Fishery Biotechnology, Chinese Academy of Fishery Sciences, Beijing 100141, China; houmingxi@cafs.ac.cn (M.H.); wangqi@cafs.ac.cn (Q.W.); zhaoran@cafs.ac.cn (R.Z.); caoyiming@cafs.ac.cn (Y.C.); zhangjin@cafs.ac.cn (J.Z.); sunxiaoqing@cafs.ac.cn (X.S.); styuwork@163.com (S.Y.); zhangyan@cafs.ac.cn (Y.Z.); 2Chinese Academy of Agricultural Sciences, Beijing 100081, China; 3National Demonstration Center for Experimental Fisheries Science Education, Shanghai Ocean University, Shanghai 201306, China; 18631836881@163.com (K.W.); cyjttkl@163.com (Y.C.)

**Keywords:** sex, transcription factors, motif enrichment, methylome, expression regulation, common carp

## Abstract

Epigenetic modifications are critical in precisely regulating gene expression. The common carp (*Cyprinus carpio*) is an economically important fish species, and females exhibit faster growth rates than males. However, the studies related to epigenetic modifications in the common carp gonads are limited. In this study, we conducted the Assay for Transposase Accessible Chromatin sequencing (ATAC-seq) and Bisulfite sequencing (BS-seq) to explore the roles of epigenetic modifications in the common carp gonads. We identified 84,207 more accessible regions and 77,922 less accessible regions in ovaries compared to testes, and some sex-biased genes showed differential chromatin accessibility in their promoter regions, such as *sox9a* and *zp3*. Motif enrichment analysis showed that transcription factors (TFs) associated with embryonic development and cell proliferation were heavily enriched in ovaries, and the TFs Foxl2 and SF1 were only identified in ovaries. We also analyzed the possible regulations between chromatin accessibility and gene expression. By BS-seq, we identified 2087 promoter differentially methylated genes (promoter-DMGs) and 5264 gene body differentially methylated genes (genebody-DMGs) in CG contexts. These genebody-DMGs were significantly enriched in the Wnt signaling pathway, TGF-beta signaling pathway, and GnRH signaling pathway, indicating that methylation in gene body regions could play an essential role in sex maintenance, just like methylation in promoter regions. Combined with transcriptomes, we revealed that the expression of *dmrtb1-like*, *spag6*, and *fels* was negatively correlated with their methylation levels in promoter regions. Our study on the epigenetic modifications of gonads contributes to elucidating the molecular mechanism of sex differentiation and sex maintenance in the common carp.

## 1. Introduction

Sex differentiation refers to the process by which undifferentiated gonads develop into ovaries or testes after sex determination [1]. Although the triggers for sex determination vary among species, ranging from genetic to environmental factors [2], the downstream genes and signaling pathways involved in sex differentiation are conserved, and their expression patterns show sexual dimorphism. For instance, *dmrt1*, *gsdf*, *amh*, and *sox9* are highly expressed in testes, while *foxl2* and *cyp19a1a* are predominantly expressed in ovaries [3,4,5]. In addition to ovary and testis development, sex differentiation is accompanied by a range of sexual dimorphisms, such as the growth rate, color, and size [6]. The mono-sex population is desirable in the aquaculture industry, as females and males exhibit differences in economic values [7].

Epigenetic modifications are momentous in sex determination and differentiation [8]. Chromatin accessibility refers to the degree of physical contact between nuclear macromolecules and DNA, of which altering is a kind of significant epigenetic modification [9]. Numerous factors, such as the conformation and composition of nucleosomes, histone architectural proteins, RNA polymerases, and transcription factors (TFs), can influence DNA accessibility [10,11]. Active regulatory elements and motifs tend to be depleted for nucleosomes and are accessible to transcription machinery and enzymes [12]. Therefore, insights from these open regions will advance our comprehension of transcription factor binding and the regulatory potential of genetic loci. The Assay for Transposase Accessible Chromatin sequencing (ATAC-seq) is a sensitive method for obtaining the open chromatin regions [13]. Currently, the roles of chromatin accessibility in fish sex differentiation have only received limited attention.

DNA methylation is the most extensively studied and understood epigenetic modification in teleost [14]. Many previous studies have demonstrated that methylation in promoter regions is critical for programming sexual fate and maintaining sexual identity by negatively regulating sex-biased gene expression [15,16,17,18]. For instance, the methylation status of *dmrt1* and *foxl2* promoter regions governs sex differentiation in Japanese flounder (*Paralichthys olivaceus*) [19,20]. DNA methylation machinery displays dynamic shifts during sex reversal in bluehead wrasse (*Thalassoma bifasciatum*) [21].

The common carp (*Cyprinus carpio*) is a crucial fish cultivated worldwide for food consumption [22]. In 2021, the global output of common carp was about 4.41 million metric tons, accounting for approximately 5.2% of the freshwater fishery production [23]. The common carp conducts the XX/XY sex-determination system, with females growing faster than males [24,25]. Breeding the all-female carp population can improve efficiency and increase the yield for aquaculture industries. The study on reproduction is significant for improving productivity and sustainable production.

Previous studies have identified many differentially expressed genes between ovaries and testes in the common carp [26]. A detailed description of gonadal histological and sex-biased gene expression during early sex differentiation was provided [27]. The deletion of *cyp17a1* results in XX common carps developing into males and being capable of normal spermatogenesis [28]. In addition, plenty of transcriptome sequencing has uncovered numerous sex-biased genes [23,29,30]. However, the function of epigenetic modifications in sex differentiation remains unknown in the common carp.

In the present study, we generated chromatin accessibility landscapes and DNA methylome profiles of the common carp gonads to understand the molecular mechanism of sex differentiation. We identified differential epigenetic modifications between ovaries and testes and explored the possible epigenetic-mediated regulation of gene expression. To the best of our knowledge, this is the first time the functions of epigenetic modifications in the common carp gonads have been investigated using ATAC-seq and BS-seq.

## 2. Results

### 2.1. Profiles of Chromatin Accessibility in Ovaries and Testes

We performed ATAC-seq to explore the chromatin accessibility of the common carp ovaries and testes. We generated 223.52 million raw reads. After quality filtering, 220.62 million clean reads were retained and mapped to the reference genome [31] (Appendix A). The quality of ATAC-seq libraries can be assessed based on the distribution of reads near transcription start sites (TSS). As shown in Figure 1A, the reads were highly enriched in TSS, implying a low background in our profiles. Pearson’s correlation heatmap indicated a high level of consistency within groups but a low consistency across groups (Figure 1B).

Peak calling was conducted with an adjusted *p* ≤ 0.05. We finally identified 120,701 peaks in the ovaries and 193,469 in the testes. We grouped these peaks into four categories based on their genome regions: the promoter (3 kb upstream of TSS), exon, intron, and intergenic region. We found that 34.32% of peaks in the ovaries and 22.13% in the testes were enriched in promoter regions, much higher than the proportion of promoter regions in the whole genome. Moreover, 37.06% of peaks in the ovaries were enriched in gene bodies and 24.44% were enriched in intergenic regions, while 41.12% of peaks in testes were enriched in gene bodies and 32.71% were enriched in intergenic regions (Figure 1C).

### 2.2. Differential Peaks between Ovaries and Testes

We performed a comparative analysis of peaks between the ovaries and testis groups. Among all the peaks, we identified 67,753 peaks unique to the ovary group and 140,521 unique to the testis group (Figure 2A; Appendix A). The reads enriched in peaks were normalized using the values of Reads Per Million mapped reads (RPM), and the RPM values of peaks in the ovaries and testes were present utilizing a scatter plot (Figure 2B). To obtain high-confidence differential peaks, we set a fold change of RPM greater than two as the threshold. Finally, we found that 84,207 peaks exhibited more enrichment in the ovaries and 77,922 peaks exhibited more enrichment in the testes. These genome regions in which differential peaks were located were regarded as differentially accessible regions (Figure 2C).

### 2.3. Functional Annotation of Chromatin Accessibility in Common Carp Gonads

The open chromatin regions around genes can predict the interaction sites between regulatory elements and genes [32]. All peaks were assigned to their nearest genes. The common carp is an allo-tetraploid species derived from recent whole genome duplication [33]. We found that 47.42% (123,881) of peaks were in the subgenome A, and 50.33% (131,482) were in the subgenome B. There were significantly enriched peaks in the promoter regions of sex-biased genes, such as *sox9a* and *zp3*. The details and annotations of all peaks are shown in Appendix A.

Genes associated with differentially accessible regions were denoted as differentially accessible genes (DAGs). In comparing ovaries to testes, more accessible regions were associated with 34,037 genes, and less accessible regions were associated with 24,999 genes. The overlapping 17,605 genes were associated with increased and decreased chromatin accessibility regions simultaneously, implying that they could be co-regulated by several regulatory elements (Appendix A). These DAGs were then subjected to enrichment analysis. The most significant enrichment GO terms were cellular, metabolic, and biosynthetic processes at the Biological Processes level, intracellular at the Cell Component level, and organic cyclic compound binding and heterocyclic compound at the Molecular Function level (Figure 3A). Although the KEGG analysis of all DAGs showed that no significant pathway was enriched, genes more accessible in testes showed significant enrichment in the TGF-beta signaling pathway, neuroactive ligand–receptor interaction, and calcium signaling pathway (Figure 3B).

### 2.4. Motif Enrichment Analysis

TFs regulate gene expression by recognizing and binding specific motifs. We analyzed TF binding motifs in the open chromatin regions to mine potential regulations, and the most enriched motifs of each sample were shown in a bubble diagram (Figure 4A). In comparing ovaries and testes, the top ten TFs enriched in ovary-biased peaks included CTCF (CCCTC-binding factor), ETS1 (E26 transformation-specific transcription factor 1), and GABPA (GA-binding protein A) (Figure 4B). These TFs were essential for embryonic development and cell proliferation. Meanwhile, the testis-biased peaks were highly enriched in TBP3 (TATA-binding protein 3), ZEB1 (Zinc finger E-Box binding homeobox 1), RXR (Retinoid X receptor), and other TFs (Figure 4C). All differential TF binding motifs between ovaries and testes are listed in Appendix A (*p* ≤ 0.01). The motifs of the well-known TFs FoxL2 (Forkhead box L2) (*p* = 1.00 × 10^−178^) and SF1 (Steroidogenic Factor 1) (*p* = 1.00 × 10^−345^), playing a pivotal role in regulating the synthesis of steroid hormones, were enriched in the ovaries (Appendix A).

### 2.5. Relationship between Chromatin Accessibility and Gene Expression

We previously conducted RNA sequencing (RNA-seq) of ovary and testis samples [34]. These samples overlapped with those in the current study, allowing us to investigate the underlying relationships between chromatin accessibility and gene expression. We divided genes into five groups based on expression levels (Figure 5A). The Fragments Per Kilobase of exon per Million mapped fragments (FPKM) values obtained from RNA-seq were used as proxies of expression levels. Genes with higher FPKM values showed stronger ATAC signals near the TSS in both the ovary and testis groups (Figure 5A). RNA-seq showed that 11,404 genes had up-regulated expression and 15,523 genes had down-regulated expression in ovaries compared to testes (*p* ≤ 0.05, log_2_foldchange > 1). We constructed a quadrant diagram to show the possible relationship between chromatin accessibility and gene expression. Red indicates positive correlations, whereas blue indicates negative correlations (Figure 5B). In ovaries compared to testes, 4745 up-regulated genes and 4339 down-regulated genes were only associated with more accessible regions, while 1087 up-regulated genes and 2558 down-regulated genes were only associated with less accessible regions. Additionally, 4072 up-regulated and 5740 down-regulated genes were simultaneously associated with increased and decreased chromatin accessibility regions (Figure 5C). We listed overlaps between DAGs and differential expression genes (DEGs) in Appendix A. We revealed that increased and decreased chromatin accessibility regions existed simultaneously near well-known genes participating in sex differentiation, such as *foxl2a*, *amh*, and *sox9a*. 

### 2.6. Summary of Bisulfite Sequencing Data

To reveal DNA methylation modifications in common carp gonads, we performed BS-seq and generated 370.75 million raw reads. After quality filtering, 363.98 million high-quality clean reads were retained. The bisulfite conversion efficiency for all libraries was over 99.71%. More than 60.56% of clean reads were mapped uniquely to the reference genome, and these reads were submitted for the next analyses. The summary of BS-seq data is shown in Appendix A. Of all genomic cytosines, the average percentages of methylated cytosines (mCs) were 3.97% and 4.07% in ovaries and testes, respectively. The methylation rates in CG contexts were 43.32% in ovaries and 44.54% in testes, while the methylation rates in CHG and CHH contexts (H represents T/A/C) were less than 0.2% in both ovaries and testes. Among all the mCs, more than 96.04% in ovaries and 96.26% in testes were mCGs (Appendix A). Methylation occurred predominantly in cytosines of CG contexts.

### 2.7. DNA Methylation Profiles of Ovaries and Testes

We calculated the methylation levels of mCGs. The results showed that most mCGs displayed high methylation levels. Especially in testes, more than 90% of the mCGs demonstrated a methylation level of over 80%. The proportion of mCGs with methylation levels over 60% in the ovaries exceeded 85% (Figure 6A). We further investigated methylation levels in different genome elements. The 5′ UTR regions showed low methylation levels, whereas the repeats and 3′ UTR regions exhibited higher methylation levels overall in both ovaries and testes (Figure 6B). The methylation levels decreased with the proximity to the TSS in 2 kb upstream regions and increased dramatically after the TSS, and the gene body regions exhibited relatively high methylation levels (Figure 6C). We subsequently analyzed differentially methylated regions (DMRs) between ovaries and testes in CG contexts (Appendix A). The number of DMRs in different genome regions was calculated. Plenty of DMRs were in gene bodies (exons and introns) and promoter regions, as shown in Figure 6D. A Circos plot was conducted to display the distribution and significance of DMRs. The DMRs were distributed relatively evenly throughout the genome (Figure 6E). There were 4392 DMRs in subgenome A and 5151 DMRs in subgenome B.

Those genes overlapping with DMRs in gene body regions or promoters were regarded as differentially methylated genes (DMGs). We finally identified 2087 promoter-DMGs and 5264 genebody-DMGs, respectively (Appendix A). Among them, 1294 genes simultaneously exhibited differentially methylated genes in the promoters and gene bodies. The KEGG enrichment analysis revealed that promoter-DMGs were enriched in metabolic pathways, pentose and glucuronate interconversions, and terms related to cytochrome P450 (Figure 7A), and genebody-DMGs were enriched in the MAPK signaling pathway, Wnt signaling pathway, TGF-beta signaling pathway, and GnRH signaling pathway (Figure 7B). Genebody-DMGs seem to be enriched in the signaling pathways related to sex differentiation. We also performed GO enrichment analysis, as shown in Appendix A.

In addition, we identified nine promoter-DMGs and 95 genebody-DMGs in CHG contexts (Appendix A) and 62 promoter-DMGs and 287 genebody-DMGs in CHH contexts (Appendix A). Venn diagrams were made to show overlapping DMGs across different contexts (Appendix A).

### 2.8. Overall Relationship between DNA Methylation and Gene Expression

We classified the genes into four groups. Genes with FPKM < 1 were considered unexpressed, and the remaining genes were divided into low- (1 ≤ FPKM < 3.76 in ovaries and 1 ≤ FPKM < 2.99 in testes), medium- (3.76 ≤ FPKM ≤ 19.08 in ovaries and 2.99 ≤ FPKM ≤ 14.20 in testes), and high (FPKM > 19.08 in ovaries and FPKM > 14.20 in testes)-expression groups based on mRNA expression levels. We then compared DNA methylation levels in CG contexts among the different groups. In both ovaries and testes, all four groups showed decreased methylation levels from the 2 kb upstream of TSS to TSS, where promoters are located. The unexpressed group showed the relatively highest methylation levels among the four groups in the promoter regions. Notably, the levels of gene expression and DNA methylation showed a strictly negative correlation in the TSS upstream and TES downstream regions in the testis group (Figure 8A).

According to the methylation levels in promoter regions, we ranked promoter-methylated genes and then equally divided them into five groups. The methylation level gates of each group were 0.17, 0.42, 0.63, and 0.76 in the ovaries and 0.16, 0.47, 0.76, and 0.91 in the testes. From the first group to the fifth group, methylation levels increased sequentially. In ovaries, less than 40% of genes were unexpressed in the first and second groups, whereas over 60% were unexpressed in the fourth and fifth groups. Similar results were observed in the testis group (Figure 8B). These results are consistent with the generally acknowledged hypothesis that promoter methylation levels are inversely correlated with transcript abundance.

### 2.9. Combined Analysis of DMGs and DEGs

A scatter plot was made to display the association between DMRs and DEGs. Red dots represent negative correlations, and blue dots represent positive correlations (Figure 8C). Finally, we identified 441 (431 + 10) promoter-hypermethylated genes whose expression levels were concurrently down-regulated and 175 (170 + 5) promoter-hypomethylated genes whose expression levels were up-regulated (Figure 8D; Appendix A). This result included many well-known genes, such as *dmrtb1-like*, *spag6*, and *fels*. Methylation in gene body regions showed more complex correlations in regulating gene expression. Overlaps between genebody-DMGs and DEGs were analyzed, and there were 1659 (837 + 130 + 74 + 618) genes whose methylation levels in gene body regions were negatively related to mRNA expression and 1671 (570 + 74 + 130 + 897) genes whose methylation levels in gene body regions were positively related to mRNA expression (Figure 8E; Appendix A).

## 3. Discussion

The common carp exhibits evident sexual dimorphism in growth, which serves as a crucial catalyst for elucidating the mechanism of sex maintenance and consequently producing mono-sex populations. Despite the extensive efforts made by scholars to uncover the distinct transcripts between ovaries and testes, there is still much room for studying the differences in epigenetic modifications.

### 3.1. Chromatin Accessibility Profiles in Common Carp Gonads

Chromatin accessibility reflects the regulatory capacity of gene expression, which is involved in various biological processes. Chromatin accessibility varies considerably between ovaries and testes [35,36]. In the common carp, our ATAC-seq data analysis identified 120,701 peaks in ovaries and 193,469 peaks in testes, respectively. A total of 34.32% peaks in ovaries and 22.13% peaks in testes were enriched in promoter regions. The proportion was significantly higher than the proportion of promoter regions in the whole genome, revealing the presence of regulatory elements in these regions [37]. The distribution of peaks was consistent with many previous ATAC-seq data and indicated the reliability of our data [38,39].

Motif analysis showed that more TFs enriched in ovaries were associated with embryonic development and cell proliferation, which might be due to the high transcriptional activity of numerous maternal genes during oogenesis [40]. The most enriched TFs in testes were systemic and related to fundamental biological processes. A previous study has found that Foxl2 and SF1 can bind to the promoter region of *cyp19a1a*, which is essential for regulating estrogen synthesis [41]. We found that motifs associated with Foxl2 and SF1 were enriched only in the ovary group, indicating that target genes of Foxl2 and SF1 were more accessible in ovaries compared to testes. Many genes were linked with multiple peaks, indicating that several regulatory elements could synergistically regulate a single gene. Integrating the analysis of ATAC-seq and RNA-seq revealed that some genes are associated with both relatively increased and decreased peaks simultaneously and that open chromatin regions could either improve or repress gene expression, which contributed to forming complex regulatory networks. Just like some well-known genes related to sex differentiation, *dmrt1* can stimulate the expression of *sox9a* and repress the expression of *foxl2*, whereas *foxl2* can inhibit *dmrt1* and *sox9a* [3,4].

### 3.2. Differential DNA Methylation between Ovaries and Testes

DNA methylation is one of the most well-understood epigenetic modifications and is crucial in various biological processes in teleost. We decoded the methylome of common carp gonads using bisulfite sequencing, presenting that over 96.04% of mCs were in CG contexts. The general methylation pattern in different genome elements was consistent with that of other fish [42,43]. For instance, methylation levels presented relatively lower degrees and decreased with proximity to TSS in the promoter regions, while the regions of repeats and 3′ UTR exhibited higher methylation levels overall.

We identified 2087 promoter-DMGs and 5264 genebody-DMGs between ovaries and testes in CG contexts. The promoter-DMGs were enriched in KEGG pathways related to cytochrome P450, which is essential for steroid hormone synthesis [44]. Several studies indicated that methylation in the gene body regions may correlate positively with gene expression or participate in alternative mRNA splicing, exhibiting a more complex mechanism in regulating gene expression [45,46]. The role of genebody methylation in sex differentiation is rarely reported. Our study showed that genebody-DMGs were enriched in the Wnt signaling pathway and GnRH signaling pathway. The Wnt signaling pathway is known to have a crucial role in ovarian differentiation and maintenance [47], and the GnRH signaling pathway is essential in modulating reproduction [48]. These findings suggest that DNA methylation, whether in promoter or genebody regions, might play a vital role in sex maintenance.

Previous studies have illustrated that methylation in the promoter regions could reversely modulate gene expression to govern sex differentiation [14]. Integrated DNA methylation and gene expression analysis revealed genes whose expressions were negatively correlated with the methylation levels in promoter regions. *dmrt1b-like* may be the mediated regulation of gene expression in this way and be involved in sex maintenance.

## 4. Materials and Methods

### 4.1. Sample Collection

The animals used in this experiment were reared at the Fangshan Experimental Station in Beijing, China. The common carp received MS222 (40 mg/L) anesthesia to alleviate their discomfort during sampling. The sexually mature common carp (18 months old) were dissected in October to obtain the gonad, which contains gametes in different development stages. The ovary and testis were subsequently frozen in liquid nitrogen for further use. The ovaries and testis surveyed in the current study had three replicates. All experiments were conducted in compliance with the principles of animal care and use for scientific purposes established by the Animal Care and Use Committee of the Chinese Academy of Fishery Sciences (protocol code ACUC-CAFS-20220615, approved on 15 June 2022).

### 4.2. ATAC Library Preparation and Sequencing

The library preparation and sequencing were conducted following previously published protocols [49]. The nucleus was extracted from digested gonad tissue and then incubated in the Tn5 transposase reaction mix. The transposase preferentially fragmented DNA in open chromatin regions, and adapter sequences were added simultaneously to the ends of the fragments. Then, PCR amplification was performed to complete the library construction. The library was sequenced on the Illumina Novaseq platform to generate 150 bp paired-end reads.

### 4.3. ATAC-Seq Reads Alignment

Adaptor sequences and low-quality bases were removed to prepare the data for downstream analysis. The clean reads were mapped to the reference genome using BWA (v0.7.12) with default parameters. The common carp reference genome and gene annotation files were downloaded from our previous study [31]. Reads aligned to mitochondrial DNA and PCR duplicates were discarded. High-quality reads (Mapping Quality ≥ 13) were used for further analysis.

### 4.4. Peak Calling and Motif Analysis

We used MACS2 (v2.1.0) to perform peak calling [50]. The dynamic Poisson distribution was employed to calculate the *p*-value of specific regions based on the uniquely mapped reads. Afterward, the Q-value (adjusted *p*-value) was derived from the *p*-value using the Benjamini–Hochberg method, and the region would be deemed as a peak when the Q-value is less than 0.05. Peaks were adjusted to the same size with summits centered, and then loci sequences were examined for motif discovery using HOMER software (v4.9.1) with default parameters [51].

### 4.5. Differential Peaks Analysis and Annotation

All alignment files were scaled to read coverage files normalized by RPM using deepTools (v3.0.2) to identify differential peaks between ovaries and testes. The differential peaks were those with a fold change of RPM greater than two (|log_2_foldchange| > 1). We retrieved gene annotation information surrounding peaks and analyzed the genomic distributions of peaks using ChIPseeker [52]. In ovaries compared to testes, the regions that enriched more/less peaks were more/less accessible regions.

### 4.6. Bisulfite Library Construction and Sequencing

Genomic DNA was extracted from gonads using the QIAamp Fast DNA Tissue Kit (Qiagen, Duesseldorf, Germany). The extracted high-quality DNA was mixed with positive control lambda DNA and then fragmented into 200–400 bp fragments using Covaris S220 (Covaris, Woburn, MA, USA). These fragments were treated with Bisulfite using an EZ DNA Methylation-Gold™ Kit (Zymo Research, Tustin, CA, USA). After bisulfite treatment, unmethylated cytosine was converted to uracil, while methylated cytosine remained unchanged. Afterward, we completed library construction using the Accel-NGS Methyl-Seq DNA Library Kit (Swift Biosciences, Ann Arbor, MI, USA). The library was then sequenced on the Novaseq platform, and 150 bp paired-end reads were generated.

### 4.7. Bisulfite Sequencing Reads Mapping

The raw data were processed to eliminate low-quality reads, adapters, and poly N sequences. The resulting clean reads were used for subsequent analysis. The reference genome and clean reads were first transformed into bisulfite-converted versions (C-to-T and G-to-A converted). The similarly converted versions of the reference genome and clean reads were aligned using Bismark software (v0.16.3) [53]. The unique and best alignment reads were then aligned back to the original reference genome to infer the methylation state of each cytosine position. The percentage of cytosines sequenced at cytosine reference positions in the lambda genome was calculated as the sodium bisulfite non-conversion rate.

### 4.8. Methylation Level Estimation and Differential Analysis

We conducted a binomial test for every cytosine site to identify methylated sites, and sites with Q-values (Benjamini–Hochberg method corrected *p*-values) less than 0.05 were thought to be methylated. The methylation level of each cytosine site was calculated as the number of reads methylated at the site divided by the total number of reads covering it, and the methylation level of a region was quantified as the average methylation level of all cytosines in this region. Differentially methylated regions between ovaries and testes were identified using DSS software (v2.12.0) [54,55].

### 4.9. Gene Functional Annotation

Enrichment analysis was conducted to aid in understanding gene functions in various biological processes using the ClusterProfiler R package (v3.9). This package included GO and KEGG pathway terms. Q-values less than 0.05 were deemed significant.

## 5. Conclusions

We generated chromatin accessibility landscapes and DNA methylomes of the common carp gonads and identified widespread alterations of epigenetic modifications between ovaries and testes. Combined with the previously published transcriptomes, we further explored the possible epigenetic-mediated regulation of gene expression. This study not only provides novel insights into the potential molecular mechanism of sex differentiation but also facilitates sex control breeding programs for the common carp.

## Figures and Tables

**Figure 1 ijms-25-00321-f001:**
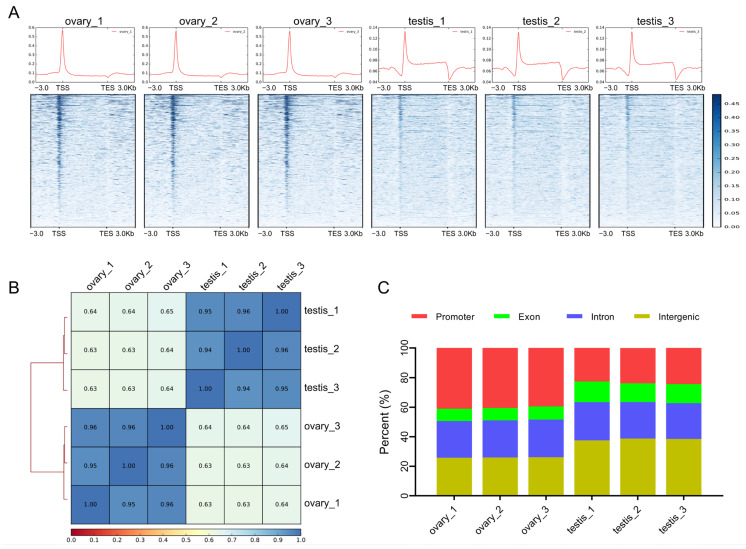
Overview of common carp gonadal chromatin accessibility profiles. (**A**) Distributions of reads around gene body regions in each sample. TSS, transcription start sites; TES, transcription end sites. (**B**) The heat map of correlations among different samples. (**C**) Proportions of peaks distributed in different genome elements.

**Figure 2 ijms-25-00321-f002:**
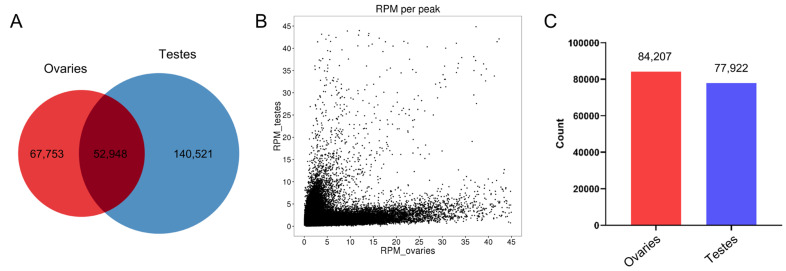
Identification of differential peaks. (**A**) The Venn diagram of peaks between ovaries and testes. (**B**) The Scatter plot of RPM for peaks in ovaries and testes. (**C**) Differential accessible regions between ovaries and testes.

**Figure 3 ijms-25-00321-f003:**
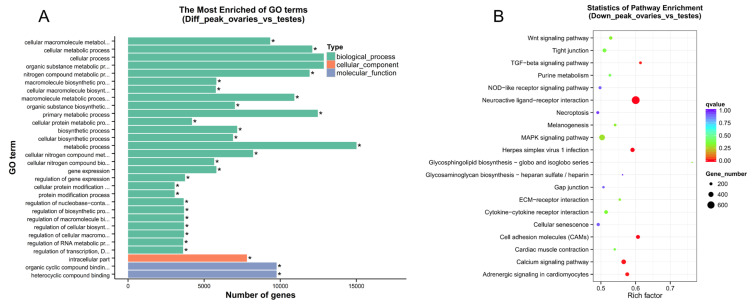
Functional annotation of differentially accessible genes. (**A**) GO enrichment analysis of DAGs between ovaries and testes. ‘*’ represents Q-values less than 0.05. (**B**) KEGG enrichment analysis of more accessible genes in the testes.

**Figure 4 ijms-25-00321-f004:**
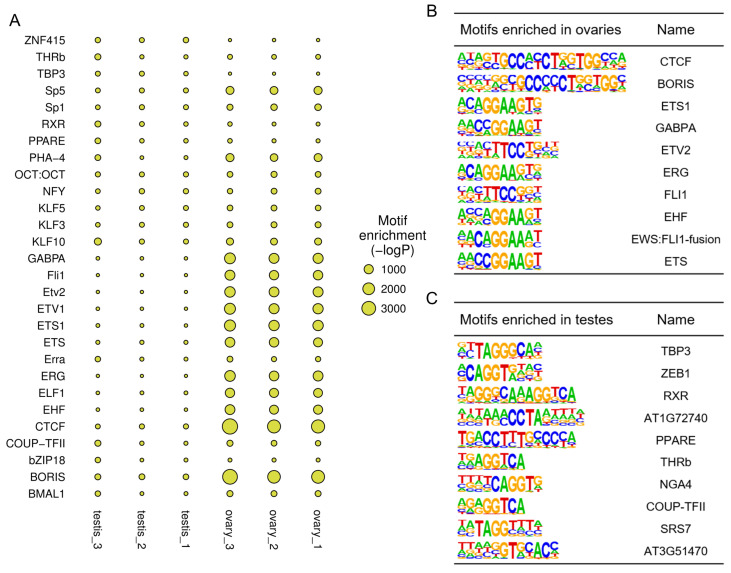
TF binding motifs enriched in open chromatin regions. (**A**) The bubble diagram of motifs enriched in each sample. (**B**) Top ten motifs enriched in ovary-biased peaks. (**C**) Top ten motifs enriched in testis-biased peaks.

**Figure 5 ijms-25-00321-f005:**
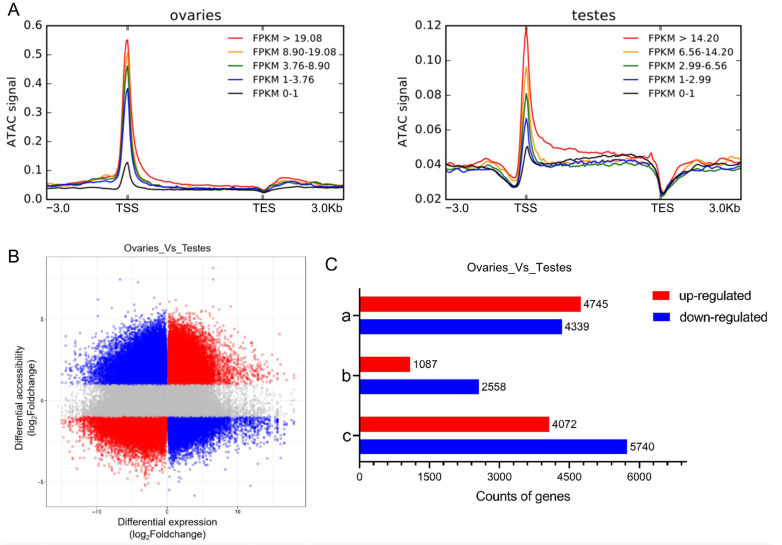
The relationship between chromatin accessibility and gene expression in common carp gonads. (**A**) The abundance of ATAC signals in gene bodies and flanking regions among gene groups with different expression levels. (**B**) The underlying relationship between chromatin accessibility and gene expression. Red dots indicate positive correlations; blue dots indicate negative correlations. (**C**) The number of overlapping genes between DAGs and differential expression genes (DEGs). Red indicates up-regulated genes; blue indicates down-regulated genes. ‘a’ represents genes only associated with more accessible regions, ‘b’ represents genes only associated with less accessible regions, and ‘c’ represents genes simultaneously associated with increased and decreased chromatin accessibility regions.

**Figure 6 ijms-25-00321-f006:**
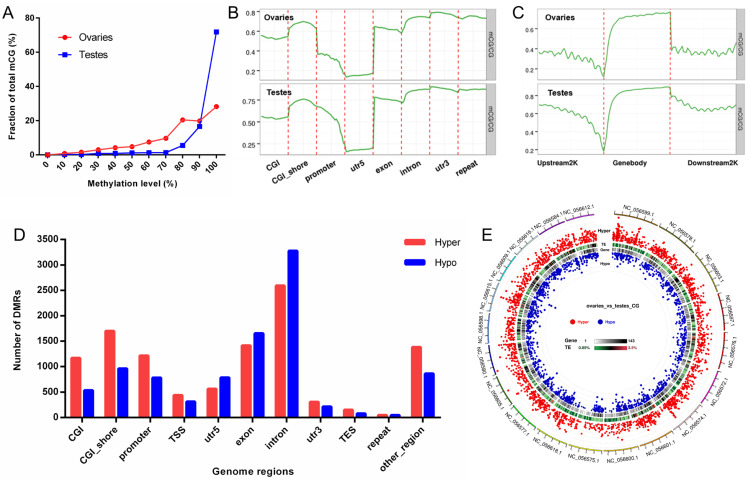
Methylation levels of mCGs in different genome regions and DMRs between the two groups. (**A**) Distribution of methylation levels in mCG contexts. (**B**) Methylation levels in different genome elements. (**C**) Methylation levels in gene bodies and flanking regions. (**D**) Number of DMRs in different genome elements. Red and blue represent hypermethylation and hypomethylation regions in ovaries compared to testes. (**E**) Distribution of DMRs in the genome. Red and blue dots represent hypermethylation and hypomethylation in ovaries compared to testes, respectively. CGI, CpG island; TSS, transcription start sites; TES, transcription end sites; TE, transposable elements.

**Figure 7 ijms-25-00321-f007:**
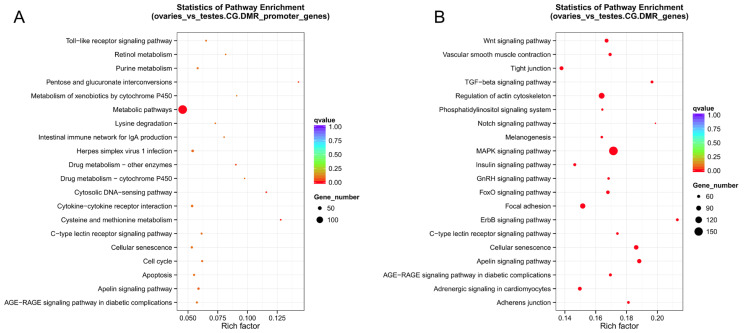
KEGG enrichment analyses of DMGs between ovaries and testes. (**A**) KEGG enrichment of promoter-DMGs. (**B**) KEGG enrichment of genebody-DMGs.

**Figure 8 ijms-25-00321-f008:**
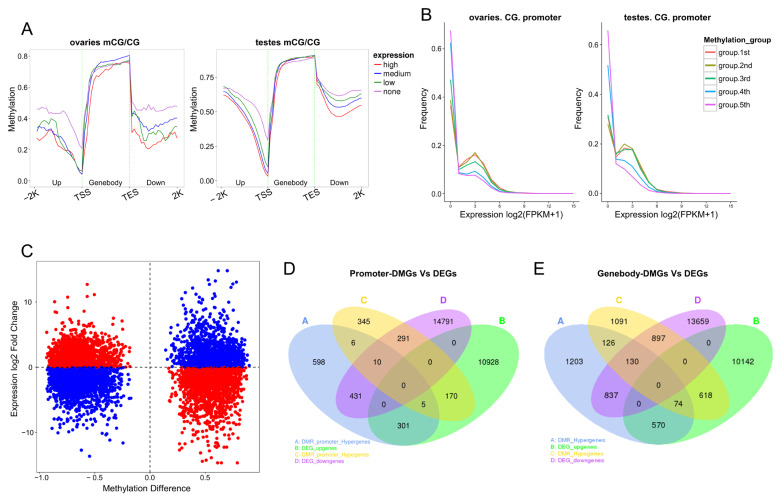
The relationship between DNA methylation and gene expression. (**A**) The methylation levels among gene groups with different expression levels. (**B**) The gene expression levels among groups with different methylation levels in promoter regions. Methylation levels increased sequentially from the first group to the fifth group. (**C**) Association analysis between DMRs and DEGs. The horizontal axis represents the methylation levels of DMRs, and the vertical axis represents gene expression levels (log_2_foldchange). Red dots represent negative correlations, and blue dots represent positive correlations. (**D**) The Venn diagram represents the overlaps of promoter-DMGs and DEGs. (**E**) The Venn diagram represents the overlaps of genebody-DMGs and DEGs.

## Data Availability

The ATAC-seq and BS-seq data presented in the current study have been submitted to the National Center for Biotechnology Information (NCBI) BioProject database under the accession number PRJNA1042881.

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
