# Peer review of "Analysis of Chromatin Accessibility and DNA Methylation to Reveal the Functions of Epigenetic Modifications in Cyprinus carpio Gonads"

_ijms, 2023, doi:10.3390/ijms25010321_

Round 1

Reviewer 1 Report

Comments and Suggestions for Authors

Overall, the study sheds light on the molecular mechanisms of sex differentiation in common carp, highlighting the significant role of epigenetic modifications in gonads. This paper focuses on the role of epigenetic modifications in the gonads of common carp (Cyprinus carpio), an economically important fish species where females grow faster than males. The study utilized Assay for Transposase Accessible Chromatin sequencing (ATAC-seq) and Bisulfite sequencing (BS-seq) to investigate these modifications.

The work is very interesting, with a well-conducted experimental design. I really liked the results and the figures presented, which greatly facilitate the understanding of the work.

Understanding the mechanisms involved may later have important practical applicability in the global economy, sustainability, and the fight against hunger.

In my opinion, the article should be accepted and I congratulate the authors on the way the work was conducted and how it is presented in the form of an article.

I just have a few suggestions:

Line 70: could you write the sentence differently?

Figure 2: it could be divided to make the text of Figure 2 D and E more readable.

Something could be added, in a more holistic and global view, regarding the importance of this work not only economically but also in productivity, fighting hunger, and the sustainability of production.

Congratulations to the authors

Reviewer 2 Report

Comments and Suggestions for Authors

Letter to Authors
ijms-2769050 -v1
Analysis of Chromatin Accessibility and DNA Methylation to Reveal the Functions of Epigenetic Modifications in Cyprinus carpio Gonads
Mingxi Hou, Qi Wang, Ran Zhao, Yiming Cao, Jin Zhang, Xiaoqing Sun, Shuangting Yu, Kaikuo Wang, Yingjie Chen, Yan Zhang, Jiongtang Li

231217

Dear authors,
You have fully characterized carp methylomes from ovary and testis for elucidating expression regulation of genes involved in sexual maturation or gametogenesis. Your cutting-edge research will contribute to understand gonochoristic fish sexual development in depth. Experimental settings and analytical methods employed are largely good except for what your materials were. Maturation stage was not given. This makes it unclear what processes the genes were involved in. Anyway, no additional data nor analysis is necessary indicating your MS can be published after a round of minor revision. See below for detail.

L25
Fox -> in Italics
SF -> in Italics
Gene names (DNAs) should be in Italics. Subtype IDs can be in Roman, though you seem Italicizing subtype IDs as well (see L32), and that is OK. Check thoroughly.

L29
Wnt -> in Italics
TGF -> in Italics
GnRH -> in Italics
"X signaling pathway" means a pathway involving a gene X.

L35 keywords
chromatin accessibility; DNA methylation; gonads; epigenetic modifications -> replace
Avoid listing words which appear also in the title. Duplicate hits upon computer search do not make sense. Give words that do not appear in the title to draw attention from wider readership. Posting words that neither appear in the abstract is better, because even in full-text search/indexing robots may not weigh much on words deeper (posterior) in the text. Hint: oogenesis, spermatogenesis, gonadogenesis, sexual maturation, motif enrichment, transcription factor, promoter, genebody, differentially accessible gene, open chromatin region, methylome, etc.

L48
a difference in economic value -> difference in economic values

L62-68
The methylation status .. a protogynous fish [21]. -> revise making compact (< 40 words)
Do not make a simple reference list (A stated this, B analyzed that, C argued it, or alike). Simple reference lists bloat your MS, dilute your originality, and undersell your own research. Use noun phrases to make abstract the reference contents. In this case particular species names other than the carp, and gene names other than Foxl2, SF1, dmrt1, spag, and fels are not essential (omit them).
-> dynamics in genomic methylation status governing differentiation and reversal of sexes was .. [refs] ?

L73,78,84,86,etc
common carp (generic) -> the common carp (specified in L70)
Check thoroughly.

L80
Zhai et al. [29] generated cyp17a1-deficient XX common carps, which developed -> cyp17a1-deficient XX common carps developed
You may cite the reference at the end.
A merit of numbered citation is to save readers' short term memory spaces when reading. Readers can go straightforward through the story-flow without outflow of author names [and published years].

L95
the reference genome
Citation is necessary. #49? If so, reference renumbering may be necessary.

L108,123,212(partly),253(partly),256 figure pictures
Fonts in the pictures are too small to see. They should at least as large as those in the main text. In addition, figure 1 and some other figure pictures seem in jpeg format. Do not compress pictures in jpeg that deceives eyes by over-smoothing with unwanted blobs. You may see how jpeg pictures obscures it by enlarging to, say, 400%.
To improve the figure picture resolution, choose either of the following ways.

The best way:
Original picture drawn by X application
 -> export to a pdf file, or copy to clipboard
 -> open it with Illustrator, or paste onto an Illustrator picture
 -> omit unwanted picture frame or canvas hemming when necessary
 -> select all
 -> slowly (important) drag-and-drop onto a word document
This makes the picture with vector data and searchable text.

2nd best:
Original picture drawn by X application
 -> export to a pdf file
 -> open it with Photoshop or GIMP upon an appropriate resolution to make it full-HD or larger size
 -> trim margins when necessary
 -> export to a png file
 -> paste it onto a word document

3rd best:
Original picture drawn by X application
 -> enlarge to full-HD or larger size on screen
 -> print-screen and paste it onto a Photoshop or GIMP picture canvas
 -> combine pictures
 -> export to a png file
 -> paste it onto a word document

IMPORTANT NOTES
You might be surprised, but PPT/XLS is hardly compatible with Word! Exact reproduction of PPT/XLS pictures in fine resolution is impossible unless mediated through pdf.
Do not use a jpeg compression option when printing to a pdf file. Use a zip compression option instead, if available.
Use print quality option when exporting a word document to a pdf file.

L120-122
Finally, 84,207 peaks .. corresponded to .. ovaries and testes ??
I cannot follow the logical structure of these statements. Enrichment and differential accessibility are in different lines of evidence. Setting a fold change point at two is arbitrary.
corresponded to ? -> were regarded as ?

L132
Common carp is an allo-tetraploid species
Reference needed.

L161
FoxL2,SF1 ?
Where are they in Figure 3? Table S4 has poor information without strength.

L219
CG island -> CpG island ?

L232
DSS software -> DSS software [54,55]
Reference renumbering may be necessary.

L235
The results revealed that (verbose) -> delete

L251
conducted (does not make sense) -> made

L267
low, medium, and high expression group -> low (< X), medium (< Y), and high (>= Y) expression group
Maybe arbitrary but the gating should be given.

L275-276
According to methylation levels ?
divided them into five groups ?
Numerical information is necessary. Equally divided into five groups from 0 (none) to X% ?
Providing numerical information here, statements in L276-279 can be compacted.

L283
Were data from the ovary and testis combined?

L284
performed (does not make sense) -> made

L286
441 ? -> 441 (431 + 10) ?

L287
175 ? -> 175 (170 + 5) ?

L290-292
Overlaps .. shown in Figure 7E and 291 Table S13. ?
How about the outline or perspective?
Wording like "X is shown in figure/table Y" imposes killing readers' times to read such an information deficient sentence telling only that there is a figure/table. You should present an outline or a perspective drawn from the figure/table and cite it in parentheses at the end.

L298,347,352
sex differentiation ?
Ovarian and testicular stages were unclear. See L358. It is unclear what stages (differentiation/maturation/gametogenesis) the genes were involved in.
Use of this term in L325,342 is currently fine, but revision would be needed according to change above.

L301-305
In orange-spotted grouper .. during the process of sex reversal [35]. (simple reference list) -> Dynamic chromatin accessibility changes associated with the processes of sex reversal in protogynous hermaphroditic fish [34,35].

L314
maternal ?
You seem talking about genomic imprinting, but is it OK?

L316
in European sea bass (Dicentrarchus labrax) -> delete or in fish

L346
suggested -> suggest

L350
616 -> delete
Exact number is not essential.

L358
Ovaries and testes of sexually mature carp
Season? How were the stages in oogenesis and spermatogenesis? Have they experienced a spawning season (at first maturity or not)?

L360-363
All experiments were .. approved on 15 June 2022). (redundant) -> delete
See L459.

L416
adjusted P-values (Benjamini-Hochberg method) -> Q-values [(Benjamini-Hochberg corrected P-values)]
See L382.

L477 references
Check the reference list carefully again from the beginning. Reference lists are frequently hotbeds of errors. You might add, omit or swap citation in the main text on the way internal revision. Numbering of the references might then shift. If so, readers think you are making irrelevant citation. It is the authors' responsibility that all references are properly cited.

Check thoroughly to make sure:
if scientific names are in Italics (L520,etc),
if journal titles are abbreviated when possible (L499,etc),
if abbreviated journal title words accompany a dot (L479,etc),
if book editors and publisher information are properly presented (L583,etc),
etc.
See the citation guide at:
https://www.mdpi.com/authors/references/

Author Response

Please see the attachment."
